# Structural insights into the competitive inhibition of the ATP-gated P2X receptor channel

Go Kasuya[1], Toshiaki Yamaura[2], Xiao-Bo Ma[3], Ryoki Nakamura[1], Mizuki Takemoto[1], Hiromitsu Nagumo[2], Eiichi Tanaka[2], Naoshi Dohmae[4], Takanori Nakane [1], Ye Yu[3], Ryuichiro Ishitani[1], Osamu Matsuzaki[5], Motoyuki Hattori[6] & Osamu Nureki[1]

P2X receptors are non-selective cation channels gated by extracellular ATP, and the P2X7 receptor subtype plays a crucial role in the immune and nervous systems. Altered expression and dysfunctions of P2X7 receptors caused by genetic deletions, mutations, and polymorphic variations have been linked to various diseases, such as rheumatoid arthritis and hypertension. Despite the availability of crystal structures of P2X receptors, the mechanism of competitive antagonist action for P2X receptors remains controversial. Here, we determine the crystal structure of the chicken P2X7 receptor in complex with the competitive P2X antagonist, TNP-ATP. The structure reveals an expanded, incompletely activated conformation of the channel, and identified the unique recognition manner of TNP-ATP, which is distinct from that observed in the previously determined human P2X3 receptor structure. A structure-based computational analysis furnishes mechanistic insights into the TNP-ATP-dependent inhibition. Our work provides structural insights into the functional mechanism of the P2X competitive antagonist.

[1] Department of Biological Sciences, Graduate School of Science, The University of Tokyo, 2-11-16 Yayoi, Bunkyo-ku, Tokyo 113-0032, Japan. [2] Laboratory for Drug Discovery, Pharmaceuticals Research Center, Asahi Kasei Pharma Corporation, 632-1 Mifuku, Izunokuni-shi, Shizuoka 410-2321, Japan. [3] Department of Pharmacology and Chemical Biology, Institute of Medical Sciences, School of Medicine, Shanghai Jiao Tong University, 280 South Chongqing Road, Shanghai 200025, China. [4] Global Research Cluster, RIKEN, 2-1 Hirosawa, Wako-shi, Saitama 351-0198, Japan. [5] Pharmaceuticals Research Center, Asahi Kasei Pharma Corporation, 632-1 Mifuku, Izunokuni-shi, Shizuoka 410-2321, Japan. [6] State Key Laboratory of Genetic Engineering, Collaborative Innovation Center of Genetics and Development, Department of Physiology and Biophysics, School of Life Sciences, Fudan University, 2005 Songhu Road, Yangpu District, Shanghai 200438, China. Go Kasuya and Toshiaki Yamaura contributed equally to this work. Correspondence and requests for materials should be addressed to O.M. (email: matsuzaki.ob@om.asahi-kasei.co.jp) or to M.H. (email: hattorim@fudan.edu.cn) or to O.N. (email: nureki@bs.s.u-tokyo.ac.jp)

ATP is the main cellular energy source and also serves as an extracellular signal transmitter[1]. P2X receptors are non-selective cation channels gated by extracellular ATP[2–4]. The vertebrate P2X receptors include seven subtypes (P2X1–P2X7) that form homo- or hetero-trimers and are involved in diverse physiological processes, such as muscle contraction, pain sensation, inflammation, and perception[5–7]. Each subunit of the P2X receptor is composed of the large extracellular domain that contains the ATP and other ligand binding sites, the two transmembrane helices that form a non-selective cation pore, and the intracellular N- and C-termini that modulate channel gating.

Among the P2X receptor subtypes, the P2X7 receptors have a unique, long intracellular C-terminus with protein binding, phosphorylation, and lipid recognition sites[8–10]. P2X7 receptors are mainly expressed in immune and nervous system cells, including macrophages, lymphocytes, neurons, and astrocytes. The activation of P2X7 receptors stimulates the release of proinflammatory cytokines, such as interleukins and tumor necrosis factor-alpha (TNF-α). Therefore, P2X7 receptors play a crucial role in inflammation, immunity, neurological function, and apoptosis[11, 12]. Accordingly, P2X7 receptors are potential therapeutic candidates for rheumatoid arthritis, hypertension, and atherosclerosis[13, 14], and clinical trials of chemical compounds targeting P2X7 have been conducted for P2X7-associated diseases[15]. In addition, the human *P2X7R* gene is located at chromosome position 12q24, and is highly polymorphic. To date, more than 600 single-nucleotide polymorphisms (SNPs) have been detected, and some of them cause amino-acid substitutions[16–18]. These substitutions induce losses or gains of functions in P2X7 receptors, and are associated with tuberculosis infection, ischemic stroke, and mood disorders[15, 16, 19].

The previously determined structures of the zebrafish P2X4 (zfP2X4; the "zf" refers to zebrafish) receptor, the Gulf Coast tick P2X (amP2X; the "am" refers to the tick's scientific name *Amblyomma maculatum*) receptor and the human P2X3 (hP2X3; the "h" refers to human) receptor provided structural insights into the recognition of agonists and antagonists, as well as the channel activation and desensitization. Quite recently, the crystal structures of the panda P2X7 (pdP2X7; the "pd" refers to panda)

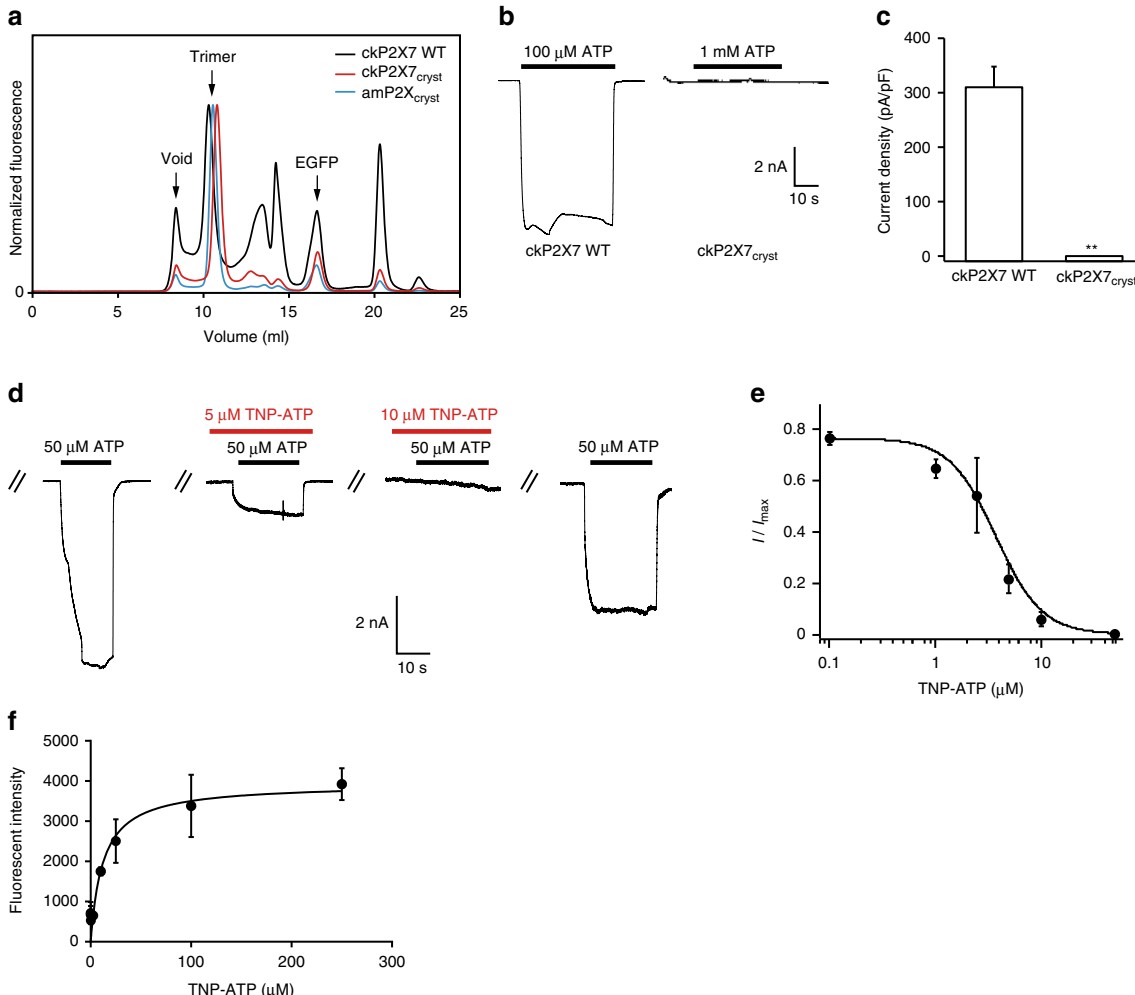

**Fig. 1** Functional properties of ckP2X7 and crystallization construct ckP2X7$_{cryst}$. **a** FSEC profiles on a Superdex 200 10/300 GL column (GE Healthcare) for the EGFP-fused ckP2X7 WT (*black*), the EGFP-fused ckP2X7$_{cryst}$ (*red*), and the EGFP-fused amP2X$_{cryst}$ (*cyan*), expressed in HEK293S GnTI$^-$ cells. The *arrows* indicate the estimated elution positions of the void volume, the EGFP-fused P2X (trimer), and the free EGFP. **b** Representative traces of ATP-evoked currents of ckP2X7 WT and ckP2X7$_{cryst}$. **c** Summarized effect (mean + s.e.m., $n = 3$–15) of the mutation on the ATP current of ckP2X7$_{cryst}$. $**p < 0.05$ vs. ckP2X7 WT. **d** Representative traces of the effects of TNP-ATP on the 50 µM ATP-evoked current of ckP2X7 WT. ATP solution was prepared under $Ca^{2+}$ free conditions. **e** The dose–response curve of TNP-ATP with ckP2X7 WT was fitted to the Hill equation (*solid line*, IC$_{50}$ = 3.55 ± 1.1 µM). Data points are means ± s.e.m. for $n = 3$. **f** Measurement of the TNP-ATP binding ability of the ckP2X7$_{cryst}$, monitored by the excitation spectrum change of TNP-ATP upon protein binding. *Error bars* indicate ±s.e.m. for $n = 5$. The calculated $K_d$ for TNP-ATP binding is 12.26 ± 2.64 µM

**Table 1 Data collection and refinement statistics**

| | TNP-ATP-bound ckP2X7 |
|---|---|
| *PDB code* | 5XW6 |
| *Data collection* | |
| Beamline | X06SA-PXI |
| Wavelength (Å) | 1.000 |
| Space group | $P4_12_12$ |
| Cell dimensions | |
| $a, b, c$ (Å) | 113.0, 113.0, 333.5 |
| $\alpha, \beta, \gamma$ (°) | 90.0, 90.0, 90.0 |
| Resolution (Å)* | 29.9–3.1 (3.23–3.10) |
| $R_{pim}$* | 0.097 (0.578) |
| $I/\sigma I$* | 5.2 (1.6) |
| Completeness (%)* | 99.8[a] (100.0) |
| Multiplicity* | 6.0 (6.1) |
| $CC_{1/2}$ (%)* | 98.6 (64.9) |
| | |
| *Refinement* | |
| Resolution (Å) | 3.1 |
| No. reflections | 40,046 |
| $R_{work}/R_{free}$[a] | 0.226/0.251 |
| No. of atoms | |
| Total | 7932 |
| Protein | 7751 |
| Ligand/others | 180 |
| Average B-factors (Å²) | |
| Protein | 75.5 |
| Ligand/ion | 74.4 |
| Water | 100.5 |
| R.m.s. deviations | |
| Bond lengths (Å) | 0.007 |
| Bond angles (°) | 0.990 |
| Ramachandran | |
| Favored (%) | 96.1 |
| Allowed (%) | 3.9 |
| Outlier (%) | 0.0 |

*Highest resolution shell is shown in parentheses
[a]About 5.0% of the reflections were excluded from the refinement for $R_{free}$ calculation

receptor provided structural insights clarifying the actions of the subtype-specific non-competitive antagonists of P2X7 receptors[20–24].

Intriguingly, the recent crystallographic and NMR analyses of P2X receptors revealed an unexpected discrepancy regarding the antagonistic mechanism by 2′,3′-O-(2,4,6-trinitrophenyl)-ATP (TNP-ATP), one of the subtype non-selective competitive P2X antagonists[25, 26]. The crystal structure of the hP2X3 receptor in complex with TNP-ATP adopted the same conformation as that in the apo, closed state[23]. In contrast, the NMR analysis of the zfP2X4 receptor reconstituted in nanodiscs showed that TNP-ATP binding induces the expansion of the extracellular domain, in a similar manner to that observed with ATP-dependent activation[27]. Therefore, the mechanism by which the competitive antagonist works at P2X receptors remains controversial.

Here, we report the crystal structure of the chicken P2X7 receptor in complex with TNP-ATP. The structural comparison with the previously determined crystal structures of P2X receptors shows an expanded, incompletely activated conformation of the channel in the chicken P2X7 receptor, and reveals the unique recognition manner of TNP-ATP, which is distinct from that observed in the hP2X3 receptor structure. A structure-based computational analysis furnishes insights into the inhibitory mechanism of TNP-ATP against P2X receptors. These findings provide structural insights into the functional mechanism of the P2X competitive antagonist.

## Results

**Functional characterization and structure determination**. We first screened the expression of P2X7 receptors and their truncation constructs by Fluorescence detection size-exclusion chromatography (FSEC)[28]. One of the truncated constructs of the chicken P2X7 receptor (ckP2X7; the "ck" refers to chicken), which shares 45.0% identity and 61.9% similarity to the human P2X7 receptor, showed a sharp, symmetrical FSEC profile, similar to that of the previously crystallized amP2X receptor (Fig. 1a)[22]. In the whole-cell patch clamp analyses using HEK-293 cells transformed with the ckP2X7 WT plasmid, ATP activated biphasic currents and TNP-ATP blocked ATP-dependent currents (Fig. 1b–e), in similar manners to other P2X7 receptors[15, 29]. The truncated expression construct of ckP2X7 (ckP2X7cryst) was designed, based on the crystallization constructs of the zfP2X4 receptors[20, 21] and the amP2X receptor[22]. The ckP2X7cryst construct lacks the regions encoding 27 N-terminal and 214 C-terminal residues, which were predicted to be structurally disordered, and includes the mutation of Asn190 to exclude a putative glycosylation site. Although we could not detect any ATP-dependent ckP2X7cryst associated current in the whole-cell patch clamp recording (Fig. 1b, c), the ckP2X7cryst construct still had TNP-ATP binding activity (Fig. 1f). Taken together, these results from the electrophysiological and fluorescent binding analyses demonstrated that the ckP2X7 receptor-associated current has similar properties to those of other P2X7 receptors and is inhibited by TNP-ATP.

The crystals of ckP2X7cryst in complex with TNP-ATP were obtained by the vapor diffusion method, and diffracted X-rays to 3.1 Å resolution. The structure of ckP2X7cryst was determined by the molecular replacement method, using the amP2X receptor structure as the search model (Table 1; Supplementary Fig. 1a, b)[22].

**Overall structure**. The overall assembly and subunit folding of ckP2X7cryst are consistent with those of the previously determined P2X structures[20–24, 30], sharing the chalice-like trimeric architecture with a large hydrophilic extracellular domain, two transmembrane helices, and intracellular termini, resembling the shape of a dolphin (Supplementary Fig. 1c)[20]. In the extracellular domain, the electron density for the TNP-ATP molecule was detected at the inter-subunit ATP binding pocket (Fig. 2a, b; Supplementary Fig. 5). While TNP-ATP acts as an antagonist of P2X7 receptors, the extracellular domain structure of ckP2X7cryst is similar to those of the ATP-bound, activated hP2X3 and zfP2X4 structures (Supplementary Fig. 8a, d), rather than those of the apo, closed hP2X3, zfP2X4, and pdP2X7 structures or the antagonist bound, closed hP2X3 and pdP2X7 structures (Supplementary Fig. 8b, c, e, f, h), consistent with the previous NMR analysis of the zfP2X4 receptor[27]. The transmembrane pore is formed by the TM2 helices with the residues from Asp319 to Leu344 lining the pore, consistent with the previous electrophysiological analysis (Fig. 2c–f)[31, 32]. In the most constricted region, the pore radius is smaller than 0.3 Å, which is too narrow for ion conduction. The constricted region is formed by the Met340 and Cys341 residues near the cytoplasmic side (Fig. 2d–f). In contrast, in the zfP2X4 and hP2X3 structures in the apo, closed state, the most constricted region is located near the extracellular side, rather than the cytoplasmic side (Supplementary Fig. 3)[20, 21, 23]. In those structures, the residues corresponding to Thr322 and Ser326 in ckP2X7 define the constricted channel gate (Supplementary Figs. 2 and 3). This partial pore opening on the extracellular side in the ckP2X7cryst structure is apparently due to the extracellular domain architecture, which is similar to those of the ATP-bound, activated P2X structures

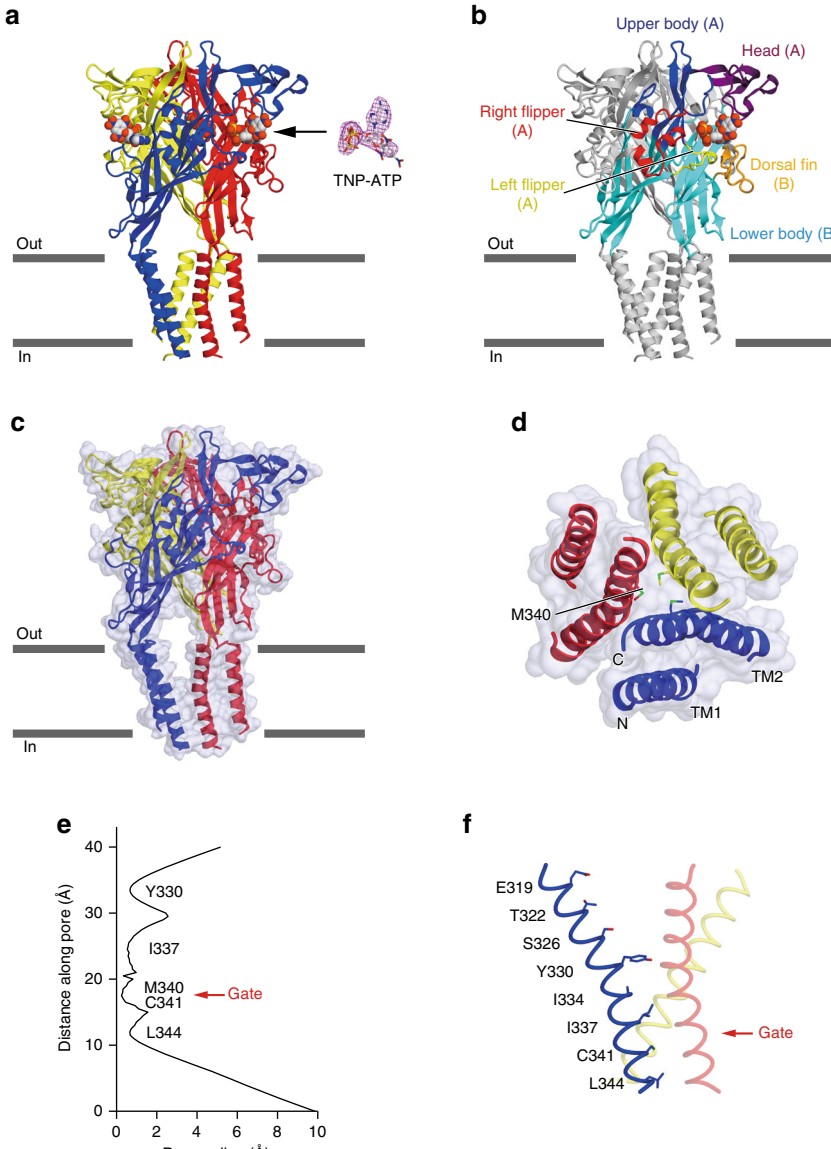

**Fig. 2** The architectures of the TNP-ATP-bound ckP2X7 structure. **a** The TNP-ATP-bound ckP2X7 structure, viewed parallel to the cell membrane. The *blue*, *red*, and *yellow* colors correspond to each subunit. The omit $F_o$–$F_c$ density map contoured at 2.5 σ is presented for the TNP-ATP molecular density. **b** The recognition manner of TNP-ATP. The molecule is colored according to the dolphin-like model[20]. **c** The surface model with a cartoon representation of the TNP-ATP-bound ckP2X7 structure. **d** The surface model with a cartoon representation of the transmembrane domain in the TNP-ATP-bound ckP2X7 structure, viewed from the intracellular side. Amino-acid residues involved in the pore constriction region are depicted by stick models. **e** The pore radius for the TNP-ATP-bound ckP2X7 structure along the pore center axis. The pore size was calculated with the program HOLE. **f** Pore-lining residues of the TNP-ATP-bound ckP2X7 structure are shown in *stick representations*

(Supplementary Fig. 8a, d). Notably, the transmembrane domain architectures of the previously determined ATP-bound structures of zfP2X4 and amP2X, including the large inter-subunit gaps, are partially distorted from those in the native lipid environment, potentially due to detergents or truncations utilized for crystallization[21, 22, 33]. However, such abnormal inter-subunit gaps are not observed in the transmembrane region of the ckP2X7cryst structure (Fig. 2c, d). Overall, this structure represents a TNP-bound, inactivated state of the P2X7 receptor.

**Mapping of functionally important mutations**. Numerous SNPs of P2X7 have been identified, and some are associated with

amino-acid substitutions implicated in several diseases, such as cancer, chronic pain, osteoporosis, and depression[16–18]. We mapped the amino-acid residues in the hP2X7 receptor that are substituted by SNPs and involved in losses or gains of functions[15, 16, 19] onto the ckP2X7cryst structure (Supplementary Fig. 4). While the gain-of-function substitutions are observed in both the extracellular and transmembrane domains, the loss-of-function substitutions are observed only in the extracellular domain. The mutations observed in the extracellular domain are not directly involved in trimer formation. One amino-acid residue, ckPhe179 (hLeu196Pro), which is associated with affective mood disorder[19], is involved in ligand recognition but does not form a direct hydrogen bond interaction (the superscripts "ck" and "h" refer to the chicken and human P2X7 receptors,

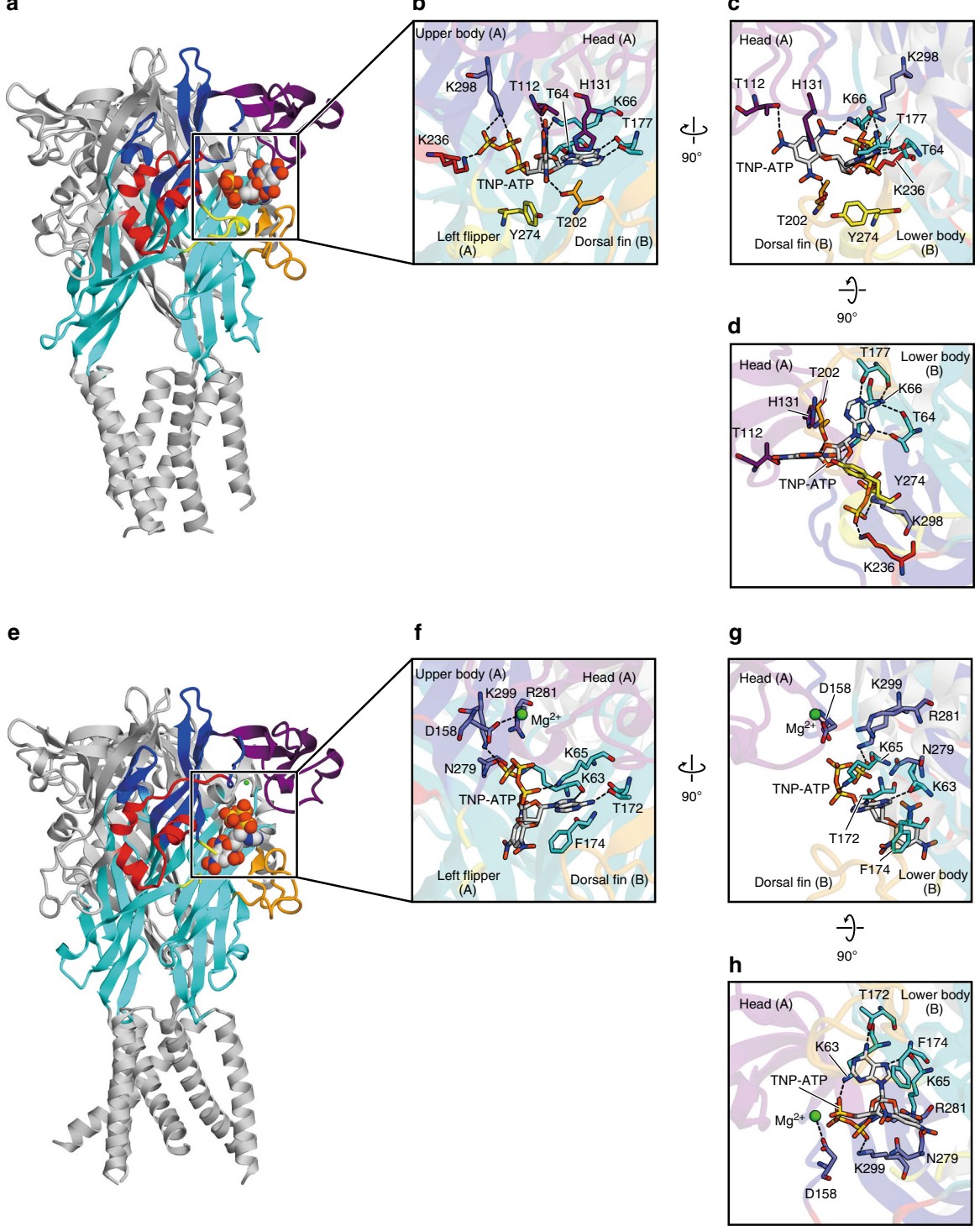

**Fig. 3** Comparison of TNP-ATP binding sites in the TNP-ATP-bound structures of ckP2X7 and hP2X3. **a–d** Overall (**a**) and close-up (**b–d**) views of the TNP-ATP binding sites in the TNP-ATP-bound ckP2X7 structure. **e–h** Overall (**e**) and close-up (**f–h**) views of the TNP-ATP binding sites in the TNP-ATP-bound hP2X3 structure (PDB ID: 5SVQ) (**b**). The molecules are colored according to the dolphin-like model[20]. The TNP-ATP and amino-acid residues are depicted by stick models. The Mg$^{2+}$ ion is depicted by a *green sphere*. *Dotted black lines* indicate hydrogen bonds (<3.3 Å)

respectively). These results indicate that the amino-acid substitutions observed in the extracellular domain are not involved in the direct trimer formation and ATP binding by the hP2X7 receptor, but they affect the conformational change of the extracellular domain required for channel activation. Intriguingly, in our inactivated ckP2X7$_{cryst}$ structure, all of the loss-of-function

substitutions are located in the secondary structure (α-helix and β-sheet) regions, suggesting that the loss-of-function substitutions may affect the structural integrity in the inactivated state. In the transmembrane domain, two substitutions, $^{ck}$Val335 ($^h$Ala348Thr) and $^{ck}$Leu344 ($^h$Thr357Ser), which are associated with toxoplasmosis and bipolar disorder[16], are observed close to

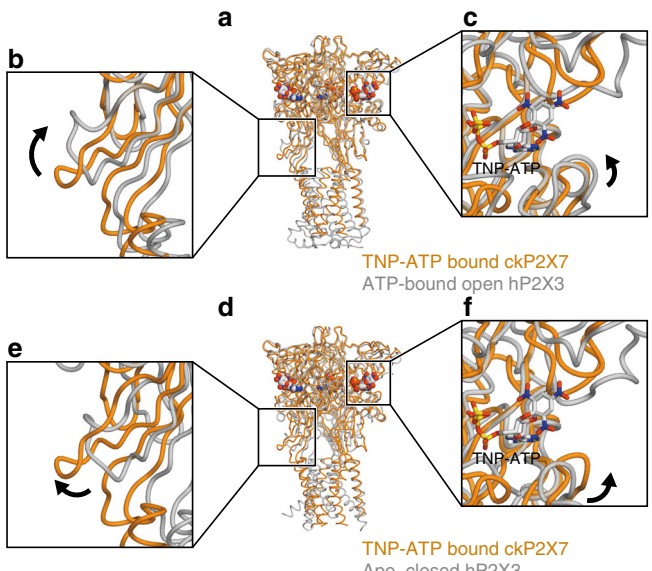

**Fig. 4** Comparison of the overall structures of the apo closed, and TNP-ATP-bound and ATP-bound, open states for ckP2X7 and hP2X3. **a**–**c** Subunit comparisons of the TNP-ATP-bound ckP2X7 (*orange*) and ATP-bound, open hP2X3 (*gray*, PDB ID: 5SVK) structures. Close-up views of the lower body domain (**b**) and the TNP-ATP binding site in the TNP-ATP-bound ckP2X7 structure (**c**) are shown in each box. The *black arrows* denote the movement from the TNP-ATP-bound state to the ATP-bound, open state. **d**–**f** Subunit comparisons of the TNP-ATP-bound ckP2X7 (*orange*) and apo, closed hP2X3 (*gray*, PDB ID: 5SVJ) structures. Close-up views of the lower body domain (**e**) and the TNP-ATP binding site in the TNP-ATP-bound ckP2X7 structure (**f**) are shown in each box. The *black arrows* denote the movement from the apo, closed state to the TNP-ATP-bound state

the pore constriction site, suggesting that these substitutions may affect the pore size of the hP2X7 receptor.

**TNP-ATP recognition**. In the TNP-ATP-bound ckP2X7 structure, the TNP-ATP molecule is located in the ATP binding pocket, and thus interacts with the head, upper body, right flipper, and left flipper domains from one subunit, and the lower body and dorsal fin domains from the neighboring subunit (Fig. 2b). Unexpectedly, the structure revealed that the binding mode of TNP-ATP is quite different from that in the recently reported TNP-ATP-bound hP2X3 structure (Fig. 3)[23].

First, the adenine ring of TNP-ATP adopts a similar orientation to that observed in the previously reported ATP-bound P2X structures[21–23], and thus forms hydrogen bonds with the side chains of the highly conserved [ck]Thr177 and the main chain carbonyl groups of [ck]Thr64 and [ck]Thr177 (Fig. 3a–d; Supplementary Figs. 2, 6). Additional hydrogen bonds are formed with the side chain of [ck]Thr64 (Fig. 3a–d; Supplementary Figs. 2, 6). In contrast, in the TNP-ATP-bound hP2X3 structure, the adenine ring of TNP-ATP adopts a distinct orientation from those observed in the ATP-bound structures (Fig. 3e–h; Supplementary Figs. 2, 6). It interacts with the side chain of [h]Thr172 ([ck]Thr177) and the main chain carbonyl group of [h]Lys63 ([ck]Thr64), but not with the main chain carbonyl group of [h]Thr172 ([ck]Thr177) (Fig. 3e–h; Supplementary Figs. 2, 6), due to the different orientation of the adenine ring.

Next, in the TNP-ATP-bound ckP2X7 structure, the phosphate groups of TNP-ATP adopt a totally extended conformation, whereas the phosphate groups of ATP in the previous structures form a bent, "U-shaped" conformation (Fig. 3a–d; Supplementary Fig. 6). This extended conformation of the phosphate groups in the TNP-ATP-bound ckP2X7 structure enables the formation of two hydrogen bonding interactions with the side chains of [ck]Lys236 from the right flipper domain and [ck]Lys298 from the upper body domain. The interactions between the right flipper domain and the phosphate groups are not visible in the previously determined ATP-bound P2X structures (Supplementary Fig. 6a–c). In contrast, the phosphate groups of TNP-ATP in the TNP-ATP-bound hP2X3 structure are partially bent in a "Y-shaped" conformation (Fig. 3e–h; Supplementary Fig. 6).

Finally, the ribose group of TNP-ATP in the TNP-ATP-bound ckP2X7 structure adopts a similar orientation to that in the ATP-bound structures, with the C2′ and C3′ atoms of the ribose group facing toward the head domain. This orientation of the ribose group and the following α-phosphate is stabilized by weak interactions with the left flipper domain at the [ck]Tyr274 residue, in a similar manner to that observed in the ATP-bound hP2X3 structure (Fig. 3a–d; Supplementary Fig. 6c, d). However, the ribose group of TNP-ATP in the TNP-ATP-bound hP2X3 structure is rotated by ~180°, and thus the C2′ and C3′ atoms of the ribose group face toward the left flipper region (Fig. 3e–h; Supplementary Fig. 6). The different orientation of the ribose group also changes that of the trinitrophenyl group: the trinitrophenyl group in the TNP-ATP-bound ckP2X7 structure contacts the head and dorsal fin domains, forming three hydrogen bonds with the side chains of [ck]Lys66, [ck]Thr112, and [ck]Thr202 (Fig. 3a–d). Consistently, previous electrophysiological analyses of the P2X1 and P2X4 receptors indicated that the head and dorsal fin domains are associated with the TNP-ATP binding[34, 35]. In contrast, the trinitrophenyl group in the TNP-ATP-bound hP2X3 structure is buried in the hydrophobic region between the lower body domain and the mostly disordered left flipper region, and thus there is no hydrophilic interaction between the receptor and the trinitrophenyl group, despite its high hydrophilicity (Fig. 3e–h; Supplementary Fig. 6).

**Putative antagonistic mechanism**. To gain insights into how TNP-ATP works at the receptor, we first compared the TNP-ATP-bound ckP2X7 structure with the apo, closed, and ATP-bound, open states of previously solved P2X receptor structures (Fig. 4; Supplementary Fig. 7). The structural comparisons between the apo, closed, and ATP-bound, open states of hP2X3 revealed that ATP binding induces the closure of the cleft between the head domain and the dorsal fin in the ATP binding pocket, and the following movement of the lower body domain to open the pore (Fig. 4). Since the extracellular domain architecture of the TNP-ATP-bound ckP2X7 structure is closer to that of the ATP-bound, open hP2X3 structure, rather than that of the apo, closed state (Fig. 4), there are similarities between the cleft closure of the ATP binding pocket and the expanded lower body domain in the TNP-ATP-bound ckP2X7 structure (Fig. 4a–c). However, these structural motifs, including the head, dorsal fin, and lower body domains in the TNP-ATP-bound ckP2X7 structure, adopt positions in between those in the apo, closed, and ATP-bound, open states of hP2X3 structures. Thus, the cleft closure of the ATP binding pocket and the expansion of the lower body domain are not completed in the TNP-ATP-bound ckP2X7 structure. Consistently, the ion-conducting pore is still closed in the TNP-ATP-bound ckP2X7 structure (Fig. 2d–f). This "incompletely

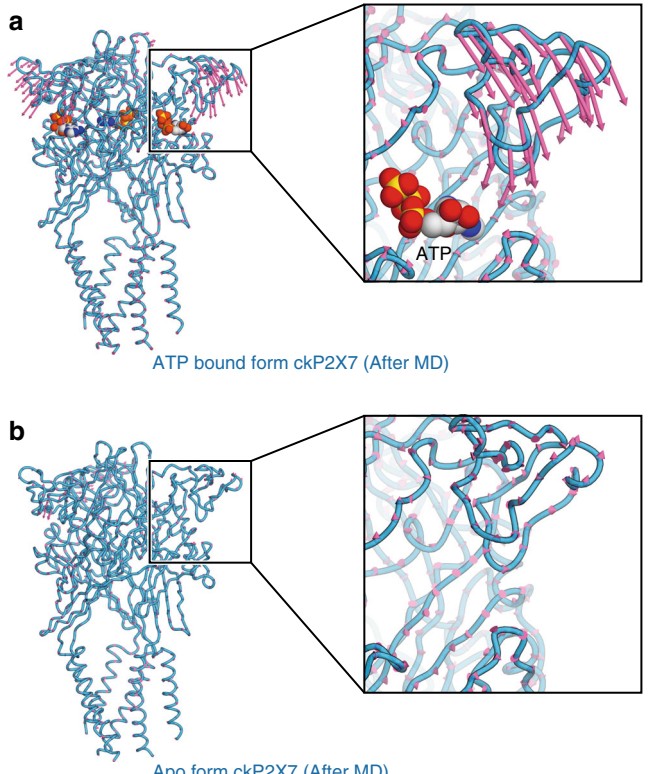

**Fig. 5** Effects of the trinitrophenyl group of TNP-ATP upon binding to P2X receptors. **a**, **b** Representative P2X7 structures during the 200 ns MD simulations in the presence of ATP (**a**) and in the absence of ligand (**b**) at the ATP binding pockets. The *pink arrows* are vector representations of the motions of the extracellular domains in the principal component analysis (PCA) of the ckP2X7$_{cryst}$

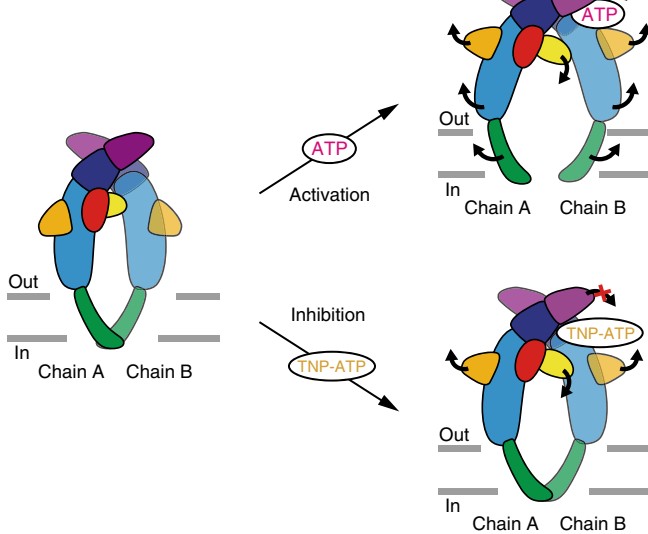

**Fig. 6** Mechanism of TNP-ATP-dependent inhibition. Cartoon models of ATP-dependent activation and TNP-ATP-dependent inhibition. The *black arrows* indicate the movements activated by ATP from the apo, closed state (*left*) to the ATP-bound, open state (*right*, *above*). The *black arrow* with a *red cross* indicates the movement caused by TNP-ATP from the apo, closed state (*left*) to the TNP-ATP-bound, closed state (*right*, *below*)

activated" conformation of the TNP-ATP-bound ckP2X7 structure is apparently due to the insertion of the trinitrophenyl group between the head and dorsal fin domains, which prevents the complete cleft closure motion in the ATP binding pocket (Fig. 4). The structural comparisons between ckP2X7 and zfP2X4 are also consistent with those between ckP2X7 and hP2X3 (Supplementary Fig. 7). To further investigate the functional role of the trinitrophenyl group upon the cleft closure motion, we performed all-atom molecular dynamics (MD) simulations of the ckP2X7$_{cryst}$ structure[22, 36]. By removing the trinitrophenyl group of TNP-ATP and all-atoms of TNP-ATP, we created the "ATP-bound form" and "Apo form" ckP2X7 structures, respectively. These two structures were embedded into the lipid bilayer. After performing 200 ns MD simulations, in the ATP-bound form ckP2X7 structure, the head domain underwent a significant downward movement toward the ATP binding pocket, while the other domains, including the dorsal fin domain involved in the adenine ring and ribose ring recognitions through hydrophobic interactions upon ATP binding, did not undergo any movements (Fig. 5a; Supplementary Fig. 6a–c). In contrast, in the apo form ckP2X7 structure, no specific structural rearrangement was observed (Fig. 5b). These results indicate that the trinitrophenyl group prevents the proper cleft closure motion for channel opening, especially by inhibiting the proper downward movement of the head domain. Consistently, the previous electrophysiological analyses showed that ATP binding and the following downward movement of the head domain toward the ATP binding pocket are crucial for the channel activation[34, 37]. Taken together, we propose that TNP-ATP

binding may induce structural changes that are similar to those associated with ATP binding, and that the trinitrophenyl group of TNP-ATP may act as a wedge to inhibit the channel activation, by preventing the downward movement of the head domain upon the cleft closure motion of the ATP binding pocket (Fig. 6).

## Discussion

In this work, we report the crystal structure of the chicken P2X7 receptor (ckP2X7) in complex with TNP-ATP, providing clarification of the TNP-ATP recognition by the P2X7 receptor, as well as precise structural interpretations of P2X7 mutations and SNPs (Figs. 2 and 3; Supplementary Fig. 4). The comparison with the previously determined P2X structures revealed that the ckP2X7 structure represents the "incompletely activated" conformation (Fig. 4; Supplementary Fig. 7). The recently determined pdP2X7 structures showed that P2X7 receptors have the subtype-specific inter-subunit cavity formed by the upper body domains between two neighboring subunits, and that ATP binding to P2X7 receptors induces the closure of this inter-subunit cavity and the following downward movement of the head domain toward the ATP binding pocket[24]. In the ckP2X7 structure, this cavity is partially closed, as compared to that in the pdP2X7 structures (Supplementary Fig. 8f–h). These results also strengthen our proposal that the ckP2X7 structure represents the "incompletely activated" conformation.

Intriguingly, the TNP-ATP-bound ckP2X7 structure exhibits a completely distinct overall conformation and TNP-ATP recognition mode from those of the recently reported TNP-ATP-bound hP2X3 structure (Fig. 3)[23]. In the structure of ckP2X7 co-crystallized with TNP-ATP, the extracellular domain architecture is similar to that in the ATP-bound, open state structures of P2X receptors[21–23], and the trinitrophenyl group of TNP-ATP faces toward the head and dorsal fin domains (Fig. 3a–d; Supplementary Fig. 6). These structural features are quite consistent with the previous electrophysiological analyses of the P2X1 and P2X4

receptors, in which the head domain is involved in the TNP-ATP recognition[34, 35], and the NMR analysis of the P2X4 receptor, in which TNP-ATP binding induces the expansion of the extracellular domain, in a similar manner to that observed with ATP-dependent activation[27]. In contrast, in the structure of hP2X3 soaked with TNP-ATP, the overall conformation is essentially identical to the apo, closed state of hP2X3 (Fig. 3e–h; Supplementary Fig. 8c)[23], and the trinitrophenyl group of TNP-ATP rotates by ~180° and faces toward the lower body and left flipper domains (Fig. 3e–h; Supplementary Fig. 6). Intriguingly, although the left flipper domain in hP2X3 faces toward the trinitrophenyl group of TNP-ATP and is close enough to interact with it, this domain is mostly disordered in the TNP-ATP-bound structure of hP2X3. Overall, these striking structural differences between the P2X7 and P2X3 receptors might arise from the differences in the subtypes or potentially from the ligand soaking employed for the determination of the TNP-ATP-bound hP2X3 structure, whereas we employed a co-crystallization method for the structure determination of the TNP-ATP-bound ckP2X7. Further functional and structural investigations will be required to fully understand these structural differences.

In summary, our TNP-ATP-bound ckP2X7 structure not only provides insights into the antagonistic mechanism of the P2X7 receptor by the competitive antagonist, but also will facilitate structure-based drug design targeting this important ion channel family associated with various immune system diseases.

## Methods

**Protein expression and purification**. The chicken P2X7 crystallization construct (ckP2X7$_{cryst}$) was prepared by using the synthesized ckP2X7 WT (NCBI Accession Number: XP_001235163) sequence (GenScript) and subcloned into the pEG BacMam vector, with an N-terminal GFP-His8 tag, a tobacco etch virus (TEV) cleavage site and following the GSGS linker (Supplementary Table 1), and was expressed in HEK293S GnTI− (N-acetylglucosaminyl-transferase I− negative) cells (ATCC, cat. no. CRL-3022)[38]. Cells were collected by centrifugation (5000×g, 10 min, 4 °C) and broken by sonication in buffer (50 mM Tris, pH 8.0, 150 mM NaCl) supplemented with 5.2 µg ml−1 aprotinin, 2 µg ml−1 leupeptin, and 1.4 µg ml−1 pepstatinA (all from Calbiochem). Cell debris was removed by centrifugation (10,000×g, 10 min, 4 °C). Membrane was collected by ultracentrifugation (138,000×g, 1 h, 4 °C). The membrane fraction was solubilized for 1 h at 4 °C in buffer (50 mM Tris, pH 8.0, 150 mM NaCl, 2 mM CaCl₂, 10% glycerol, 2% n-dodecyl-β-D-maltoside (DDM) (Calbiochem), and 0.2 Unit l−1 Apyrase (New England Biolabs)). Insoluble materials were removed by ultracentrifugation (138,000×g, 1 h, 4 °C), and the supernatant was incubated with Talon metal affinity resin (Clontech), washed with 20 mM imidazole and eluted with 250 mM imidazole. After TEV protease digestion and Endo H treatment, the fraction was purified by size-exclusion chromatography on a Superdex 200 Increase 10/300 GL column (GE Healthcare), equilibrated with SEC buffer (20 mM HEPES, pH 7.0, 100 mM NaCl, 10% glycerol, 0.05% DDM). The peak fractions of the protein were collected and concentrated to 3 mg ml−1, using a centrifugal filter unit (Merck Millipore, 30 kDa molecular weight cutoff).

**Crystallization and data collection**. Before crystallization, 0.1 mM TNP-ATP (TOCRIS) was added to the protein solutions. The crystals were grown at 4 °C by the sitting drop vapor diffusion method in a 96-well plate. The crystals of ckP2X7$_{cryst}$ appeared within 3 days under the following conditions (0.5 M sodium sulfate, 0.05 M lithium chloride, 0.05 M Tris, pH 8.0, 32% PEG400). After cryo-protection in reservoir solution supplemented with 25% glycerol, and flash-cooling in liquid nitrogen, the crystals were used for X-ray diffraction experiments. The X-ray diffraction data were collected at the Swiss Light Source (SLS) beamline X06SA-PXI, processed using DIALS[39], and scaled using AIMLESS[40].

**Structure determination**. The TNP-ATP-bound ckP2X7$_{cryst}$ structure was obtained by molecular replacement with MOLREP[41], using one subunit of the ATP-bound amP2X structure from the Gulf Coast tick (PDB: 5F1C) as the template. The structure thus obtained was further refined by using REFMAC5[42], PHENIX[43], and COOT[44], with three-fold non-crystallographic symmetry and secondary structure restraints. Data collection and refinement statistics are summarized in Table 1.

**Electrophysiology**. ATP and TNP-ATP were purchased from Sigma (St. Louis, MO, USA). The ckP2X7 WT and ckP2X7$_{cryst}$ constructs were subcloned into pCDNA3.1 vector, and transfections of these plasmids were performed using Hilymax (Dojindo Laboratories, Kumamoto, Japan). Both constructs were expressed in HEK-293 cells, cultured in DMEM medium at 37 °C in a humidified atmosphere of 5% CO₂ and 95% air. Electrophysiological measurements were performed on HEK-293 cells, 24–48 h after transfection. The conventional whole-cell configuration under the voltage clamp at room temperature (23 ± 2 °C) was used to obtain the electrophysiological recordings[36, 45]. Patch pipettes were pulled from glass capillaries, using the two-stage puller PP-830 (Narishige Co., Ltd.), and the resistance between the recording electrode filled with pipette solution and the reference electrode in bath solution ranged from 3 to 5 MΩ. Membrane currents were filtered at 2 kHz using a low-pass Bessel Filter, and measured with an Axon 200B patch clamp amplifier (Molecular Devices). All currents were sampled and analyzed in the Digidata 1440 interface, using Clampex and the Clamp-fit 10.0 software (Molecular Devices). Cells were incubated in bath solution, containing 150 mM NaCl, 5 mM KCl, 10 mM glucose, 10 mM HEPES, 2 mM CaCl₂, and 1 mM MgCl₂, at the conditional neutral pH of 7.35–7.40. Patch electrodes were filled with a standard internal solution, containing 30 mM NaCl, 120 mM KCl, 1 mM MgCl₂, 0.1 mM CaCl₂, and 5 mM EGTA, at the conditional neutral pH of 7.35–7.40. During electrophysiological recordings, 80–90% of the series resistance was compensated and the recording electrode was held at −60 mV throughout the experiment. ATP solutions were prepared in the batch buffer within 2 h of use, and applied using a fast pressure-driven, computer-controlled microperfusion system, OctaFlow08P (ALA Scientific Instruments)[45]. ATP currents were normalized to the cell membrane capacitance. Dose–response curve data were collected from recordings of a range of TNP-ATP concentrations. The corresponding currents were normalized to the maximal current amplitude of ATP (50 µM in Ca²⁺-free solution containing 5 mM EGTA), and ATP-gated currents were recorded after the regular 40–50 s ATP application every 8 min.

**Fluorescence measurement**. The fluorescence measurement using TNP-ATP as an indicator was performed based on the previous experiment[46]. The purified ckP2X7$_{cryst}$ protein and the SEC buffer mixed with each concentration of TNP-ATP were incubated at 4 °C for over 1 h, respectively. The mixed solutions were dispensed in a 96-well, flat-bottom plate, in a 50 µl sample volume. Fluorescence measurement was performed using a 2030 ARVO X3 Multilabel Reader (PerkinElmer), equipped with 405 nm emission and 535 nm excitation filters.

**Molecular dynamics simulations**. The simulation system was comprised of the ckP2X7$_{cryst}$ trimer, 1-palmitoyl-2-oleoyl-phosphatidylcholine (POPC), ATP, water molecules, and 150 mM NaCl. The missing atoms, such as hydrogens in the protein, were added with the programs GROMACS 5.1.4[47] and VMD 1.9.3[48]. The disordered side chains in the protein were modeled by COOT[44]. The periodic boundary system, including the explicit solvent and the POPC lipid bilayer, was prepared. The protein was embedded into the POPC lipid bilayer using the MemProtMD pipeline[49]. The final size of the simulation box was 119 × 119 × 144 Å. The net charge of the system was neutralized by adding chloride and sodium ions. The topologies and force field parameters from CHARMM36[50] were used. MD simulations were performed with the program GROMACS 5.1.4[47]. The system was first minimized using the steepest descent with a cutoff of 100.0 kJ mol−1 nm−1, with 1000.0 kJ mol−1 nm−2 harmonic position restraints for non-hydrogen atoms of the protein and ATP. Next, we performed an equilibration for 100 ps in the NVT ensemble (310 K, 119 × 119 × 144 Å volume), followed by an equilibration for 6 ns in the NPT ensemble (310 K, 1 atm), with the same restraints. Finally, we performed the production runs in the NPT ensemble for 200 ns, with the following bond length restraints between the atoms of ATP and the protein side chains: (1) ATP N1 and Thr177 OG1, (2) ATP N6 and Thr177 OG1, (3) ATP N6 and Thr64 O, and (4) ATP N7 and Thr64 OG1. These restrained bond lengths were based on those observed in the ckP2X7$_{cryst}$ crystal structure. The simulation was repeated five times, and we obtained similar results. Constant temperature was maintained by the Nose–Hoover thermostat[51, 52]. Constant pressure was maintained by the Parrinello–Rahman barostat[53]. Long-range electrostatic interactions were calculated using the particle mesh Ewald method[54]. The LINCS algorithm was used for bond constraint[55]. Principle component analysis was conducted by in-house scripts using Python 2.7.9 and MDAnalysis 0.15.0[56].

**Data availability**. Atomic coordinates and structure factors for the TNP-ATP-bound ckP2X7$_{cryst}$ structure have been deposited in the Protein Data Bank (http://www.pdb.org), under the accession code 5XW6. Other supporting data are available from the corresponding authors upon reasonable request.

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

## Acknowledgements

We thank the members of the Nureki lab, especially Tomohiro Nishizawa and Yongchan Lee, for assistance with structure determination, and the beamline staff members at X06SA-PXI of Swiss Light Source (Villigen, Switzerland) for technical assistance during data collection. This work was supported by grants from the Ministry of Science and Technology of China (Grant Nos. 2016YFA0502800 to M.H., and 2014CB910302 to Y.Y.); grants from the National Natural Science Foundation of China (Grant Nos. 31570838 and 31650110469 to M.H., and 31570832 to Y.Y.); grants from the National Thousand Young Talents award from the Office of Global Experts Recruitment in China to M.H., grants from the AMED-CREST Program "The Creation of Basic Medical Technologies to Clarify and Control the Mechanisms Underlying Chronic Inflammation" to O.N.; grants from the Platform for Drug Discovery, Informatics and Structural Life Science from the Ministry of Education, Culture, Sports, Science and Technology (MEXT) to O.N.; grants from JSPS KAKENHI (Grant Nos. 17J06101 to G.K., and 16H06294 and 24227004 to O.N.); and grants from PRESTO, JST to M.H.

## Author contributions

G.K. and T.Y. prepared the constructs and performed the compound screening for crystallization, with assistance from N.D., T.Y., H.N., E.T. and O.M. X.-B.M. and Y.Y. performed the electrophysiological analysis. G.K. expressed, purified, and crystallized

the protein, and determined the structure with assistance from T.N., R.I. and M.H. R.N., M.T. and R.I. performed the molecular dynamics simulation. G.K., M.H. and O.N. wrote the manuscript. O.M., M.H. and O.N. supervised all of the research.

## Additional information

**Competing interests:** The authors declare no competing financial interests.

