## [Peer Review file · Nature Communications]

Reviewers' comments:

Reviewer #1 (Remarks to the Author):

The manuscript describes the X-ray structure of the chicken P2X7 receptor, an ATP-activated ion channel receptor, in complex with a nucleotide antagonist, TNP-ATP (a moderately potent competitive antagonist). The P2X7 receptor is of importance as a potential drug target in immunology, inflammation and cancer.

The structure has a moderate resolution of 3.1 Angstrom.

The most interesting finding is the fact that the obtained structure differs from that obtained with the human P2X3 receptor in complex with TNP-ATP, and the conformation of the inactive receptor resembles the activated conformation of the P2X4 receptor. A mechanism is provided for inhibition of the P2X7 receptor by TNP-ATP.

Major points:

1. The sequence homology between the investigated chicken and the human P2X7 receptor is only moderate. Nevertheless, the ATP binding site is similar. A truncated sequence has been crystallized.
2. Only a single compound - the weak antagonist TNP-ATP - has been co-crystallized. It would considerably strengthen the study if an additional co-crystal structure with the agonist BzATP would have been obtained, and both structures could have been compared. It would also be interesting to co-crystallize more potent (nanomolar) competitive antagonists.
3. Since TNP-ATP is such a weak antagonist of P2X7 receptors it might show different binding modes in different crystal structures. Have the authors obtained only one structure?
4. The resolution is moderate and some domains near the orthosteric binding site are not well resolved.
5. Figure 1b: Not enough data points in the steep part of the curve.
6. Figure 1c: What concentration of BzATP was used? The antagonist curve is not good and should be repeated several times. It does not start at 1.0 (100%) and it does not go down to 0.
7. Did the authors find any contribution of the loop region (left flipper) and the dorsal fin for binding / binding orientation of TNP-ATP? This should be discussed and visualized.
8. Explain and discuss the interaction of K236 (from the right flipper) with the phosphate group. This interaction is not found in other crystal structures. Is this because of the U-shaped vs. extended conformation? Or is there a difference in the conformation of the right flipper?

Minor points:

- Explanations of abbreviations are provided too late, e.g. ck, h on p. 10. Abbreviations have been explained when they appear for the first time.

Reviewer #2 (Remarks to the Author):

The manuscript describes the crystal structure of chicken P2X7, a purinergic ligand-gated anion channel, in complex with an inhibitor (TNP-ATP). They show that the conformational state of the protein in the presence of TNP-ATP has an open extracellular domain similar to the ATP-bound state, though the transmembrane pore is closed. Their results agree with an NMR study investigating the binding of TNP-ATP to P2X4 but differ from the crystal structure of human P2X3 with the same ligand. In the latter, the conformation of the protein resembles the apo protein. The manuscript is reasonably well written and illustrated and is therefore easy to follow.

Comments:

My initial impression from this manuscript was that the study was clearing up a controversy in how TNP-ATP binds to and antagonises P2X receptors. However, this is clearly not the case. The affinity of TNP-ATP to P2X7 (this study) and P2X4 (NMR study) is 1000-fold less than to P2X3 (previously published crystal structure). Thus, it is not unreasonable to expect the ligand to bind in different ways to the respective receptors. Though the authors acknowledge that different binding modes may be caused by different sub-types, they don't discuss the relative affinities and almost lead the reader to believe that the differences are due to the soaking in the previous experiments (which of course, could be true). Overall the study is interesting because it adds information about how antagonists bind and inactivate this important class of proteins but it doesn't go beyond this.

The structure of P2X7 from panda was recently solved. Though this is mentioned there is no discussion of similarities and differences between the structures.

The authors map the positions on the structure of naturally occurring mutations of the protein that lead to disease, however, they don't discuss them with respect to any mechanistic insight.

The construct that is used for the crystal structure is inactive – presumably because it is not trafficked properly, though this is not explicitly stated. The affinity for the ligand should be measured in some other way.

There is very scant information about the crystal structure and the methodology used. What is the CC0.5? How was the refinement done - NCS, secondary structure restraints etc? How many outliers in the Ramachandran plot? What were the B-factors of the ligand with respect to the other atoms?

Minor comments

Figure 3 would be easier to interpret if the figures in the top and bottom panels were in similar orientations.

#1 comments: Kasuya et al. NCOMMS-16-29362

“Reviewers' comments: Reviewer #1 (Remarks to the Author):

The manuscript describes the X-ray structure of the chicken P2X7 receptor, an ATP-activated ion channel receptor, in complex with a nucleotide antagonist, TNP-ATP (a moderately potent competitive antagonist). The P2X7 receptor is of importance as a potential drug target in immunology, inflammation and cancer.

The structure has a moderate resolution of 3.1 Angstrom.

The most interesting finding is the fact that the obtained structure differs from that obtained with the human P2X3 receptor in complex with TNP-ATP, and the conformation of the inactive receptor resembles the activated conformation of the P2X4 receptor. A mechanism is provided for inhibition of the P2X7 receptor by TNP-ATP.”

We appreciate the positive comment regarding our manuscript. The specific points mentioned by reviewer #1 are addressed below.

“Major points:

1. The sequence homology between the investigated chicken and the human P2X7 receptor is only moderate. Nevertheless, the ATP binding site is similar. A truncated sequence has been crystallized.”

We thank the reviewer for this comment. All of the previously determined P2X receptors have similar overall trimeric structures. Furthermore, as we showed in **Supplementary Fig. 6**, all of the structures in complex with ATP, including zfP2X4, amP2X and hP2X3, have similar ATP binding sites, and their amino acid residues related to ATP binding are highly conserved throughout the P2X receptor family. Therefore, we think that the ATP binding site in the human P2X7 receptor is similar to that in the chicken P2X7 receptor.

Using a truncated and/or mutated sequence is a common strategy for structural studies, to remove disordered regions and to reduce the heterogeneity of the target protein. Actually, the crystallization constructs of zfP2X4, amP2X, hP2X3 and pdP2X7 all have truncations and mutations.

“2. Only a single compound - the weak antagonist TNP-ATP - has been co-crystallized. It would considerably strengthen the study if an additional co-crystal structure with the agonist BzATP would have been obtained, and both structures could have been compared. It would also be interesting to co-crystallize more potent (nanomolar) competitive antagonists.”

We thank the reviewer for this comment. Based on the reviewer's suggestion, we co-crystallized the ckP2X7_{cryst} with the high affinity agonist BzATP. However, we could not obtain crystals with good diffraction quality. Furthermore, we tried to co-crystallize the ckP2X7_{cryst} with the other antagonists with micromolar affinity, since there are no known antagonists with nanomolar affinities to the ckP2X7 receptor. However, we still could not obtain well-diffracting crystals in this experiment. Alternatively, to gain further mechanistic insights, we performed an all atom Molecular Dynamics (MD) simulation of the ckP2X7_{cryst} structure. By removing the trinitrophenyl group of TNP-ATP and all-atoms of TNP-ATP, we created the “ATP-bound form” and “apo form” ckP2X7 structures, respectively. After the MD simulation, we observed that the head domain underwent a significant downward movement in the ATP-bound form of the ckP2X7 structure, while we did not observe any specific structural rearrangement in the apo form of the ckP2X7 structure. These results indicate that the trinitrophenyl group prevents the proper cleft closure motion for channel opening, by inhibiting the proper movement of the head domain, rather than that of the dorsal fin domain. We believe that these results strengthen our discussion. We added the corresponding descriptions in the “Putative antagonistic mechanism” section, in the revised manuscript (Page 14, line 10 to Page 15, line 6).

“3. Since TNP-ATP is such a weak antagonist of P2X7 receptors it might show different binding modes in different crystal structures. Have the authors obtained only one structure?”

As mentioned above, we also tried to cocrystallize ckP2X7_{cryst} with TNP-ATP under different crystallization conditions, but so far we have obtained only one structure.

“4. The resolution is moderate and some domains near the orthosteric binding site are not well resolved.”

Thank you for the comment, and we apologize for our poor explanation. In **supplementary Fig. 5**, we showed the F_o-F_c and $2F_o-F_c$ maps of TNP-ATP, as well as the $2F_o-F_c$ map, of the side chains of amino acid residues associated with TNP-ATP binding. These figures revealed that the TNP-ATP and the residues associated with TNP-ATP binding are clearly resolved. Overall, we agree that the resolution of the ckP2X7 structure is moderate, but the quality of our structure is still sufficient to support our conclusion that the binding mode of the TNP moiety in the ckP2X7 structure is quite different from that in the hP2X3 structure, with the $\sim 180^\circ$ degree rotation of the trinitrophenyl group of the TNP-ATP.

“5. Figure 1b: Not enough data points in the the steep part of the curve.”

Thank you for the comment. We agreed with this comment and removed the **Fig. 1b** data.

“6. Figure 1c: What concentration of BzATP was used? The antagonist curve is not good and should be repeated several times. It does not start at 1.0 (100%) and it does not go down to 0.”

We agreed with this comment and removed the **Fig. 1c** data. Alternatively, we performed the fluorescent binding assay to validate the binding activity of the ckP2X7_{cryst} to TNP-ATP (**Fig. 1f**).

“7. Did the authors find any contribution of the loop region (left flipper) and the dorsal fin for binding / binding orientation of TNP-ATP? This should be discussed and visualized.”

Thank you for the comment. In the revised manuscript, we stated that the ^{ck}Y274 residue in the left flipper domain is involved in stabilizing the orientation of the ribose and the following α -phosphate groups with weak interactions, in a similar manner to the stabilization observed in the ATP-bound hP2X3 structure. Furthermore, we also stated that the ^{ck}T202 residue in the dorsal fin domain contacts the trinitrophenyl group of the TNP-ATP, by forming a hydrogen bond interaction (Page 12, lines 5-8. 11-14).

“8. Explain and discuss the interaction of K236 (from the right flipper) with the phosphate group. This interaction is not found in other crystal structures. Is this because of the U-shaped vs. extended conformation? Or is there a difference in the conformation of the right flipper?”

Thank you for the comment. As reviewer #1 pointed out, we think that the interaction of ^{ck}K236 (from the right flipper) with the phosphate group in our ckP2X7 structure is enabled due to the extended conformation of the phosphate group in TNP-ATP. We added the corresponding descriptions in the revised manuscript (Page 11, lines 14-17).

“Minor points:

- Explanations of abbreviations are provided too late, e.g. ck, h on p. 10. Abbreviations have been explained when they appear for the first time.”

Thank you for the comment. Based on this comment, we explained the abbreviations of species names (zf, am, h, and pd) on page. 4 and ck on page. 6, where they appeared for the first time.

Reviewer #2 comments: Kasuya et al. NCOMMS-16-29362

“Reviewer #2 (Remarks to the Author):

The manuscript describes the crystal structure of chicken P2X7, a purinergic ligand-gated anion channel, in complex with an inhibitor (TNP-ATP). They show that the conformational state of the protein in the presence of TNP-ATP has an open extracellular domain similar to the ATP-bound state, though the transmembrane pore is closed. Their results agree with an NMR study investigating the binding of TNP-ATP to P2X4 but differ from the crystal structure of human P2X3 with the same ligand. In the latter, the conformation of the protein resembles the apo protein. The manuscript is reasonably well written and illustrated and is therefore easy to follow. ”

We appreciate the positive comment regarding our manuscript. The specific points mentioned by reviewer #2 are addressed below.

“Comments:

My initial impression from this manuscript was that the study was clearing up a controversy in how TNP-ATP binds to and antagonises P2X receptors. However, this is clearly not the case. The affinity of TNP-ATP to P2X7 (this study) and P2X4 (NMR study) is 1000-fold less than to P2X3 (previously published crystal structure). Thus, it is not unreasonable to expect the ligand to bind in different ways to the respective receptors. Though the authors acknowledge that different binding modes may be caused by different sub-types, they don't discuss the relative affinities and almost lead the reader to believe that the differences are due to the soaking in the previous experiments (which of course, could be true). Overall the study is interesting because it adds information about how antagonists bind and inactivate this important class of proteins but it doesn't go beyond this.”

We appreciate this comment. According to this comment, we toned down the description in the “Discussion” section. Particularly, we deleted the descriptions that the ligand soaking method hindered the TNP-ATP-dependent structural change of hP2X3. Furthermore, we described that further functional and structural investigations would be required to understand the structural differences between the ckP2X7 and hP2X3 structures (Page 18, lines 1-2).

“The structure of P2X7 from panda was recently solved. Though this is mentioned there is no discussion of similarities and differences between the structures.”

Thank you for the comment. Since the structural data of pdP2X7 receptors were not released in the Protein Data Bank when we submitted this manuscript, we could not mention the similarities and differences between the ckP2X7 and pdP2X7 structures at that time. Based on this comment, we compared our ckP2X7 structure with the pdP2X7 structure.

The pdP2X7 structures revealed that P2X7 receptors have the subtype-specific inter-subunit cavity, and this inter-subunit cavity closes for channel activation upon ATP-binding. In our ckP2X7 structure, this cavity is partially closed as compared to the pdP2X7 structures, which also strengthens our proposal that the ckP2X7 structure represents the “incompletely activated” conformation. We added the corresponding figures (**Supplementary Fig. 8**) and descriptions in the “Results” and “Discussion” sections in the revised manuscript (Page 7, line 15 to Page 8, line 3; Page 16, lines 7-15)

“The authors map the positions on the structure of naturally occurring mutations of the protein that lead to disease, however, they don’t discuss them with respect to any mechanistic insight.”

Thank you for the comment. Based on this comment, we reexamined the effects of SNPs in our P2X7 structure, and found that all of the loss-of-function SNPs are located in the secondary structure regions. This result implies that these substitutions affect the structural integrity in the inactivated state and act as loss-of-function substitutions. We added the corresponding descriptions in the revised manuscript (Page 10, lines 2-5).

“The construct that is used for the crystal structure is inactive – presumably because it is not trafficked properly, though this is not explicitly stated. The affinity for the ligand should be measured in some other way.”

Based on the comment, we performed the fluorescent binding assay to validate the binding activity of ckP2X7_{cryst} to TNP-ATP (**Fig. 1f**). We added the corresponding descriptions in the revised manuscript (Page 6, line 15 to Page 7, line 1).

“There is very scant information about the crystal structure and the methodology used. What is the CC0.5? How was the refinement done - NCS, secondary structure restraints etc? How many outliers in the Ramachandran plot? What were the B-factors of the ligand with respect to the other atoms?”

Thank you for the comment, and we apologize for our poor explanation. We added the CC1/2 and B-factors to **Supplementary Table 1**. Furthermore, upon structure determination, since we used one subunit of the previously determined ATP-bound ampP2X structure (PDB: 5F1C) as the search template in the molecular replacement, and performed further refinement with restraints of the three-fold non-crystallographic symmetry (NCS) and the secondary structure, we added these explanations in the Methods of the “Structure determination” section for a clearer explanation (Page 20, lines 11-16).

“Minor comments

Figure 3 would be easier to interpret if the figures in the top and bottom panels were in similar orientations.”

Thank you for the comment. Based on the suggestion, we changed the previous **Fig. 3** and presented the overall and close-up views of the TNP-ATP binding sites of the ckP2X7 and hP2X3 structures in similar orientations, to facilitate an easier understanding of the differences between the two structures.

REVIEWERS' COMMENTS:

Reviewer #1 (Remarks to the Author):

The manuscript has been improved and could be accepted in the present form.

Reviewer #2 (Remarks to the Author):

The comments seem to have been addressed satisfactorily.